# Internet Gaming Disorder in Children and Adolescents with Attention Deficit Hyperactivity Disorder

**DOI:** 10.3390/children9030428

**Published:** 2022-03-18

**Authors:** Stefano Berloffa, Andrea Salvati, Giulia D’Acunto, Pamela Fantozzi, Emanuela Inguaggiato, Francesca Lenzi, Annarita Milone, Pietro Muratori, Chiara Pfanner, Federica Ricci, Laura Ruglioni, Annalisa Tacchi, Chiara Tessa, Arianna Villafranca, Gabriele Masi

**Affiliations:** 1IRCCS Stella Maris, Scientific Institute of Child Neurology and Psychiatry, viale del Tirreno, 331A, Calambrone, 56025 Pisa, Italy; stefano.berloffa@fsm.unipi.it (S.B.); giulia.dacunto@fsm.unipi.it (G.D.); pamela.fantozzi@fsm.unipi.it (P.F.); emanuela.inguaggiato@fsm.unipi.it (E.I.); francesca.lenzi@fsm.unipi.it (F.L.); annarita.milone@fsm.unipi.it (A.M.); pietro.muratori@fsm.unipi.it (P.M.); chiara.pfanner@fsm.unipi.it (C.P.); federica.ricci@fsm.unipi.it (F.R.); laura.ruglioni@fsm.unipi.it (L.R.); annalisa.tacchi@fsm.unipi.it (A.T.); arianna.villafranca@fsm.unipi.it (A.V.); 2Department of Clinical and Experimental Medicine, University of Pisa, 56128 Pisa, Italy; andrea.salvati@fsm.unipi.it (A.S.); chiara.tessa18@gmail.com (C.T.)

**Keywords:** Internet Gaming Disorder, ADHD, behavioral addiction

## Abstract

Although Attention Deficit Hyperactivity Disorder (ADHD) has been related to an increased risk for behavioral addictions, the relationship between ADHD and Internet Gaming Disorder (IGD) is still debated. The aim of this study is to address this topic by exploring the prevalence of IGD in a consecutive sample of ADHD youth, compared to a normal control group, and by assessing selected psychopathological and cognitive features in ADHD patients with and without IGD. One hundred and eight patients with ADHD (mean age 11.7 ± 2.6 years, 96 males) and 147 normal controls (NC) (mean age 13.9 ± 3.0 years, 114 males) were included in the study and received structured measures for IGD. In the ADHD group, 44% of the sample were above the IGD cut-off, compared to 9.5% in the NC group. ADHD patients with IGD presented with greater severity and impairment, more severe ADHD symptomatology, more internalizing symptoms, particularly withdrawal/depression and socialization problems, and more prominence of addiction and evasion dimensions. A binary logistic regression showed that the degree of inattention presented a greater weight in determining IGD. These findings may be helpful for identifying, among ADHD patients, those at higher risk for developing a superimposed IGD.

## 1. Introduction 

The increasing diffusion of Internet use in youth has determined, in the last decade, a high rate of potentially addictive behaviors, with negative implications on social and psychological functioning [1] and academic performance [2,3]. These negative consequences were firstly included in the clinical category of Internet Use Disorders, with related diagnostic criteria and measures, namely a diagnostic questionnaire for addiction to the Internet, the Internet Addiction Test (IAT) [4]. The diagnostic criteria proposed by the American Psychiatric Association followed the addiction model postulated by Griffiths [5,6], including salience, mood modification, tolerance, withdrawal, conflict, and relapse [7]

More recently, a specific characterization led to the new category of Internet Gaming Disorder (IGD), included in the fifth edition of the Diagnostic and Statistical Manual of Mental Disorders (DSM-5), under Section 3 (“Emerging Measures and Models”) [7]. Similarly, the World Health Organization (WHO) included Gaming Disorder in the new International Classification of Diseases, 11th Edition (ICD-11) [8]. A specific diagnostic measure, based on all nine DSM-5 criteria for IGD, is the Internet Gaming Disorder Scale–Short-Form (IGDS9-SF) [9]. However, other Internet contents, such as social networks, Internet gambling, and sexual Internet sites, which are part of the broader field of Internet Addiction Disorder, are not included in the IGD criteria [10,11]. 

There are two conceptualizations of IGD, not necessarily alternative, either as a primary addictive disorder or an epiphenomenon of other psychiatric conditions. To date, available empirical evidence suggests that IGD is prevalently associated with various mental disorders, particularly with Depression, Anxiety, Obsessive-Compulsive Disorder, Social Anxiety Disorder, and Attention Deficit Hyperactive Disorder (ADHD) [12]. However, few studies have examined associations between IGD and specific psychiatric diagnoses in referred adolescents [13,14]. 

Regarding ADHD, a neurodevelopmental disorder characterized by impulsivity, inattention, and hyperactivity, evidence from the literature suggests that this disorder may be a significant predictor of Substance Use Disorder [15] and behavioral addictions [16], which share neuropsychological features with ADHD [17,18]. A recent review shows support for the association between the severity of ADHD symptoms and the severity of IGD, even if the rates of this association are uncertain [19]. Furthermore, ADHD comorbidity may negatively affect IGD course, with changes in ADHD symptoms longitudinally associated with changes in IGD symptoms [20]. Possible mediators of this association have been explored, including affective disorders [21] and Autism Spectrum Disorders [22]. Abnormalities in the brain network related to inhibitory function or sensory integration have been reported in Korean IGD patients with ADHD [23]. Medication for ADHD may be effective in ameliorating IGD symptomatology [19]. A recent meta-analysis [24] showed that IGD patients presented gray-matter volume abnormalities in the putamen, ADHD patients showed the same in the orbitofrontal cortex, and both the disorders shared abnormalities in the prefrontal cortex. With a functional MRI, both ADHD and IGD presented abnormal activation in the anterior cingulate cortex, insular, and striatum, while only IGD presented a specific increased activation in the precuneus [24].

The aims of this study are the following: (a) To explore the prevalence of IGD in Italian drug-naïve youth with ADHD with respect to a normal control group); (b) to compare two reliable measures (IGDS9-SF and IAT) for the diagnosis of IGD in ADHD patients; and (c) to describe a possible psychopathological and cognitive profile of youth with ADHD and IGD, according to IGDS9-SF or IAT, compared to patients with ADHD without IGD.

## 2. Materials and Methods

### 2.1. Sample and Recruitment 

The clinical group, consisting of patients diagnosed with ADHD, was recruited between December 2020 and May 2021 in a third-level hospital of child and adolescent neurology and psychiatry. The participants were 108 drug-naïve children and adolescents, aged between 8 and 18 years (mean age 11.7 ± 2.6 years), prevalently males (*N* = 96, 89%). The diagnoses were based on the DSM-4 diagnostic criteria, according to the historical information and a structured clinical interview, the Schedule for Affective Disorders and Schizophrenia for School-Age Children-Present and Lifetime Version (K-SADS-PL) [25]. The clinical interview was administered by trained child psychiatrists and revised by senior child psychiatrists. The exclusion criteria were comorbidity with an intellectual disability or psychotic disorder (due to the inability to complete the questionnaires) and co-occurring psychoactive medications (to exclude possible drug effects). Regarding the power of our clinical sample size, we used G*Power (post-hoc analysis) and settled on an ES of 0.40. Our clinical sample size has a power of 0.98 for our ANOVAs and 0.99 for our regression model.

A normal control group was recruited in Italy from December 2020 to May 2021 in a northern region (Tuscany) and southern region (Campania). The sample included 147 subjects, 114 males (78%), aged between 8 and 18 years (mean age 13.9 ± 3.0 years).

### 2.2. Procedures 

All the participants (clinical and control groups) completed both the Internet Addiction Test (IAT) [4] and the Internet Gaming Disorder Scale-Short Form (IGDS9-SF) [9] questionnaire. All participants received a detailed explanation of each item, verbally for children and through written instructions for older participants.

The clinical sample was also assessed according to the clinical severity with the Clinical Global Impression-Severity score (CGI-S) [26], and according to the overall adaptive functioning with the Children Global Assessment Scales (C-GAS) [27]. Parents of ADHD patients completed the Conners’ Parent Rating Scale—Revised: Short Form (CPRS—R:S) [28]. The Child Behavior Checklist (CBCL) [29] was also administered to parents for a dimensional assessment of psychopathology. Only participants aged between 11 and 18 also completed the Use, Abuse, Dependence to Internet (UADI) [30], for a qualitative assessment of Internet addiction. Finally, the clinical group was assessed with the Wechsler Intelligence Scale for Children—Fourth Edition (WISC-IV) [31].

The study conformed to the Declaration of Helsinki. Patients and parents received detailed information on the characteristics of the assessment instruments and treatment options, and all parents gave informed written consent. The methodology of the study was approved by the Regional Ethics Committee for Clinical Trials of Tuscany (Date 27 July 2021, Number 202/2021). 

### 2.3. Measures

The Internet Addiction Test (IAT) [4], validated in an Italian version [32], consists of 20 questions, all measured on a 5-point Likert scale (score 1 for the answer “rarely” and 5 for the answer “always”). Summative scores ranging from 20 to 49 were considered normal, while scores between 50 to 79 were firstly associated with occasional to frequent problems, and scores ranging from 80 to 100 were associated with significant problems due to their Internet usage. It has recently been proposed by Kimberly Young, the developer of IAT, that the threshold of 80 may be excessively high for the identification of adolescents with internet addiction, and a cut-off point of 50 has been proposed as clinically meaningful [31]. We considered, consistently with a previous study, patients scoring above 50 as having an Internet addiction (the Cronbach alpha for this measure was 0.81 in the current sample) [27]. 

The Internet Gaming Disorder Scale–Short-Form (IGDS9-SF) [9] is a unidimensional tool including nine items reflecting all nine criteria for IGD as in the DSM-5. The IGDS9-SF is widely used in research on IGD and is supported by several cross-cultural psychometric studies [32]. The IGDS9-SF evaluates the severity of IGD on the basis of both online and offline gaming habits in the last twelve months, with a clinical cut-off of 21 in the Italian Version [33]. In the current sample, the Cronbach alpha for this measure was 0.76.

The Child Behavior Checklist for Ages 6–18 (CBCL) [29], a 118-item scale, was completed by parents to assess the behavior of children and adolescents aged 6 to 18 years, with 8 different syndromes scales (anxiety/depression, withdrawal/depression, somatic symptoms, socialization problems, thought problems, attention problems, rule-breaking behaviors, aggressive behavior), a Total Problem score, and two broad-band scores designated as Internalizing Problems and Externalizing Problems. Each item is evaluated with a 3-level Likert scale, where 0 represents “not true”; 1 is “partly or sometimes true”; and 2 is “very true or often true”.

The Conners’ Parent Rating Scale—Revised: Short Form [28] is a 27-item screening tool for children and adolescents aged 3 to 17 years that, in addition to the 12 DSM-IV criteria for ADHD (cognitive and inattention problems and hyperactivity symptoms), also evaluates behaviors suggestive for Oppositional Defiant Disorder. The parent must assign to each item a score from 0 to 3 (relative to the frequency of the problems). In the current sample, the mean of Cronbach alpha values for Conners’ scales is 0.82.

Use, Abuse, Dependence to Internet (UADI) [30] is a validated instrument for the Internet addiction classification for adolescents and young adults, with five prevalent dimensions: Dissociation (tendency to alienate from reality); impact on real life (consequences of internet use on everyday life); experimentation (the use of the internet as a testing ground for self- and emotional experimentation); dependence-addiction (including behaviors and/or symptoms of dependence such as tolerance, withdrawal, and compulsiveness); escape (use of the Internet as a strategy for escaping from daily difficulties). This measure has been previously used to explore Internet use and abuse in a psychiatric population [34]. In the current sample, the mean of Cronbach alpha values for the UADI scales is 0.82.

Wechsler Intelligence Scale for Children—Fourth Edition (WISC-IV) [29] is a standardized measure including fifteen subtests, providing a Full-Scale Intelligence Quotient (FSIQ) and four Composite or Index Scores, the Verbal Comprehension Index, the Perceptual Reasoning Index, the Working Memory Index, and the Processing Speed Index [31].

#### Statistical Analysis 

Skewness and Kurtosis were calculated to verify the normality of the distribution of data. For all variables, the skew values were <2 and kurtosis values < 3. These values indicated a normal distribution for the current data [35].

The power of the sample size was calculated using G*Power (post-hoc analysis), and we settled on an ES of 0.40 and found a power > 0.90 for all analyses.

Statistical analyses were performed with parametric tests (univariate ANOVA test) and non-parametric tests (non-parametric test χ^2^), depending on the types of variables considered (quantitative, qualitative); Cohen’s ES was calculated for variables with significant differences between groups. Finally, the characteristics associated with ADHD and IGD were explored using binary logistic regression models, using the IAT or IGDS9-SF cut-off as dependent variable and variables that had significant differences in the means as independent variables. Tests to see if the data met the assumption of collinearity indicated that multicollinearity was not a concern: Tolerance values ranged between 0.43 and 0.79, and variance inflation factors were between 1.21 and 1.78. Statistical analyses were carried out using the SPSS 19.0 program for Windows. The level of significance adopted is 0.05 (*p* < 0.05).

## 3. Results

### 3.1. Internet Addiction and Internet Gaming Disorder in ADHD vs. Control Group 

According to the IAT score, among the 108 ADHD patients assessed with the IAT test, 45 (42.1%) scored higher than 50, while in the control group, only 30 out 147 subjects (20.4%) presented a score ≥ 50 (two-way test χ^2^ 13.946, *p* < 0.001). Similarly, according to the IGDS9-SF questionnaire, 48 subjects in the ADHD group (44.4%), compared to 14 (9.5%) in the normal control group (two-way test χ^2^ 41,257, *p* < 0.001), were above the cut-off.

### 3.2. Comparison between ADHD Patients with IAT Score ≥ or <50 

Patients with higher IAT scores showed greater inattention (Conners t-score inattention; *p* < 0.001; ES = 0.84), hyperactivity (Conners t-score hyperactivity, *p* = 0.007; ES = 0.54), and ADHD index (Conners ADHD Index; *p* < 0.001; ES = 0.82). They also presented higher scores on CBCL internalizing problems (*p* = 0.016; ES = 0.47) and on the following CBCL syndrome subscales: Withdrawal/depression (*p* = 0.013; ES = 0.49), socialization problems (*p* = 0.001; ES = 0.66), attention problems (*p* = 0.004, ES = 0.57), and rule-breaking behavior (*p* = 0.023; ES = 0.38) [Figure 1]. Of note, neither functional impairment, assessed by C-GAS, nor clinical severity, according to the CGI-S, differed between groups. Similarly, WISC-IV subscales failed to show significant differences between groups. 

Means and standard deviations are reported in Table 1. 

According to the UADI, ADHD patients with IAT higher than 50 scored higher on the “experimentation” subscale (42.7 ± 11.4 vs. 32.8 ± 9.6, *p* < 0.001, ES = 0.95), “addiction” subscale (52.8 ± 7.7 vs. 38.6 ± 10.8, *p* < 0.001, ES = 1.48), “escape” subscale (51.2 ± 11.0 vs. 33.9 ± 11.5, *p* < 0.001, ES = 1.26), “dissociation” subscale (41.8 ± 10.8 vs. 27.9 ± 10.2, *p* < 0.001, ES = 1.33), and, to a lesser degree, on the “impact” subscale (48.0 ± 6.6 vs. 42.5 ± 14.8, *p* = 0.041, ES = 0.46) [Figure 2].

### 3.3. Comparison between ADHD Patients with IGDS9-SF Score ≥ 21 or <21

ADHD patients with an IGDS9-SF score above the cut-off presented greater functional impairment according to the C-GAS (*p* = 0.001; ES = 0.65), greater clinical severity on the CGI-S (*p* = 0.001; ES = 0.63), and higher scores in inattention (Conners’ inattention t-score; *p* < 0.001; ES = 0.71), hyperactivity (Conners’ hyperactivity t-score; *p* < 0.001; ES = 0.75), and ADHD Index score (Conners’ ADHD Index (*p* < 0.001; ES = 0.70). Furthermore, they scored higher in both CBCL internalizing (*p* = 0.012; ES = 0.49) and externalizing (*p* = 0.006; ES = 0.51) problems, as well in the following syndromic CBCL subscales: Withdrawal/depression (*p* = 0.009; ES = 0.53), socialization problems (*p* = 0.000; ES = 0.69), aggressive behavior (*p* = 0.045; ES = 0.39), anxious/depressed (*p* = 0.022; ES = 0.45), attention problems (*p* = 0.015; ES = 0.48), thought problems (*p* = 0.022; ES = 0.46), and rule-breaking behavior (*p* = 0.001; ES = 0.71) (Figure 3). No differences were found at the WISC-IV scores, with only lower processing speed approaching statistical significance (*p* = 0.058; ES = 0.38). Means and standard deviations are reported in Table 2. 

A comparison of patients with IGDS9-SF scores above and below the cut-off according to the UADI subscales showed that ADHD patients with higher scores also presented higher scores in the UADI “experimentation” subscale (42.6 ± 11.8 vs. 32.7 ± 9.4, *p* < 0.001, ES = 0.94), “addiction” subscale (51.8 ± 8.8 vs. 42.6 ± 10.9, *p* < 0.001, ES = 0.92), “evasion” subscale (51.2 ± 11.0 vs. 33.5 ± 11.2, *p* < 0.001, ES = 1.52), and “dissociation” subscale (42.5 ± 10.7 vs. 27.0 ± 9.2, *p* < 0.001, ES = 1.57), while the “impact” subscale (47.7 ± 6.8 vs. 42.6 ± 14.9, *p* = 0.058, ES = 0.42) only approached clinical significance [Figure 3].

### 3.4. Predictive Analysis

Two binary logistic regression models were tested, using the IAT and the IGDS9-SF cut-offs as dependent variables and variables that had significant differences in the means (see results reported above) as independent variables. The degree of inattention (Conners’ inattention subscale) was the only variable associated with ADHD and IAT conditions (*p* = 0.005), while no variables were associated with ADHD and IGDS9-SF conditions.

## 4. Discussion

Internet addiction as a whole, and more specifically Internet Gaming Disorder (IGD), is a relatively new nosographic entity, frequently associated with other psychiatric disorders. ADHD has been considered a clinical disorder possibly associated with a higher risk of developing an IGD, but clinical features of ADHD patients with or without IGD are less explored. Our aim was to compare rates of IGD in a consecutive sample of referred drug-naïve ADHD patients and a normal control group, using structured measures. Furthermore, we aimed to compare ADHD patients with or without IGD, in order to explore putative clinical features in ADHD patients at higher risk for IGD. 

Consistently with other reports [19], our study supports the hypothesis that ADHD patients have a higher risk of IGD, as they present a more than two-fold higher rate of an IAT score above the cut-off and a more than a four-fold higher rate of the IGDS9-SF score above the cut-off. Core features of ADHD may explain the association between this clinical disorder and IGD, namely the impulsive need for rapid satisfaction, as well as the tendency toward sensation-seeking behaviors. This close relationship between core features of ADHD and IGD may support a beneficial effect of ADHD pharmacological treatments to prevent or improve the IGD [36].

In order to better clarify this association, we explored clinical and cognitive features of ADHD patients presenting with IAT or IGCS9-ST scores above the cut-offs. Overall, features of ADHD patients scoring above the cut-off at the IAT and IGDS9-SF are largely overlapping. Both present more severe ADHD, according to hyperactive/impulsive and inattentive components, based on Conners’ subscales and the CBCL “inattentive” subscale. This finding may further support the above-mentioned hypothesis that more severe core symptoms of ADHD increase the risk for IGD [19]. 

Interestingly, ADHD patients with IGD also show higher CBCL internalizing problems, including anxiety/depression, withdrawal, and socialization problems. Of note, all the UADI dimensions are highly represented in patients with both higher IAT and IGDS9-SF scores. This finding suggests that ADHD patients more prone to escape from the real world and dissociate from reality, as well as those with addictive and sensation-seeking tendencies according to the UADI, may be at higher risk for developing IGD. However, the two profiles (“escape from reality” and “sensation seeking”) may discriminate different phenotypes of ADHD plus IGD patients. More specifically, ADHD patients with CBCL internalizing problems may use Internet addiction as a way of escaping from daily difficulties deriving from low-esteem and social anxieties. 

Of note, no differences in the WISC-IV profile were found between ADHD with or without higher scores on the two measures, suggesting that, notwithstanding the lower working memory and processing speed indices, this cognitive profile of our ADHD patients fails to influence the proneness to IGD. 

Regarding the differences between patients with either IAT or IGDS9-SF scores above the cut-off, only in patients with a higher IGDS9-SF score were clinical severity and functional impairment higher, compared to patients below the cut-off. Furthermore, they presented both higher CBCL externalizing and externalizing problems, particularly aggression, rule-breaking, and thought problems. Of note, our data from the predictive analysis shows that inattention, but not hyperactivity and impulsivity, more closely predicts a higher IAT, but not IGDS9-SF, score. This result may suggest that the type of stimuli provided by electronic games (for example, the fast succession of stimuli) not only interferes with the attentional deficits of ADHD, but on the contrary, may fit with these attentional deficits. 

Similarities between the two groups with either IAT or IGDS9-SF scores above the cut-off suggest that patients with severe ADHD, namely with internalizing problems (anxiety, depression, and socialization problems), and with higher UADI scores in all dimensions may be at higher risk of developing IGD. Discrepancies between the two groups may suggest that IGDS9-SF characterizes a more severe group of patients, with a “mixed” (internalizing and externalizing) presentation. 

Our previous study [34] showed that adolescents with Internet addiction clustered into four, fairly balanced, groups, according to the CBCL: (1) Low levels of both internalizing and externalizing problems (the less-severely impaired patients); (2) high levels of internalizing problems (the “Internalized patients”); (3) highest levels of both internalizing and externalizing Problems (the “mixed”, and most-severely impaired patients); and (4) high levels of externalizing problems (“the Externalized patients”). There were no significant relationships among specific clusters and Internet addiction, but the last two clusters (mixed and externalized) were more closely related to the experimentation dimension of UADI. 

Our findings in ADHD patients are consistent with these previous data, supporting the notion that even in ADHD, IGD patients may represent a heterogeneous group. Patients presenting externalizing problems, with prominent aggression and rule-breaking behaviors, may represent a specific phenotype of ADHD plus IGD patients, with impulsivity and sensation-seeking, while the presence of only internalizing problems may discriminate isolated and fearful patients, with low self-esteem and poor social strategies. Further studies with larger samples may support this distinction, which may have relevant implications for implementing personalized therapeutic approaches. 

Our findings should be interpreted in light of several limitations, namely the small sample size and the self-report questionnaires describing the Internet and Gaming addiction, as well as the UADI dimensions. The paucity of females in the clinical sample did not allow for closer comprehension of possible gender specificities, which certainly deserve specific research. Finally, we did not explore the possible effect of pharmacological treatment with stimulants in the affecting rates, severity, and course of IGD [19,36,37]. This limitation may be a possible development of the research, exploring the hypothesis that pharmacological treatment may improve not only ADHD symptoms but also IGD. Despite these limitations, our findings may be helpful for clinicians managing ADHD patients, providing suggestions for diagnostic assessment, the ascertainment of patients at higher risk, and finally, the definition of different phenotypes within the broad category of IGD, with different therapeutic needs, useful for precision medicine.

## Figures and Tables

**Figure 1 children-09-00428-f001:**
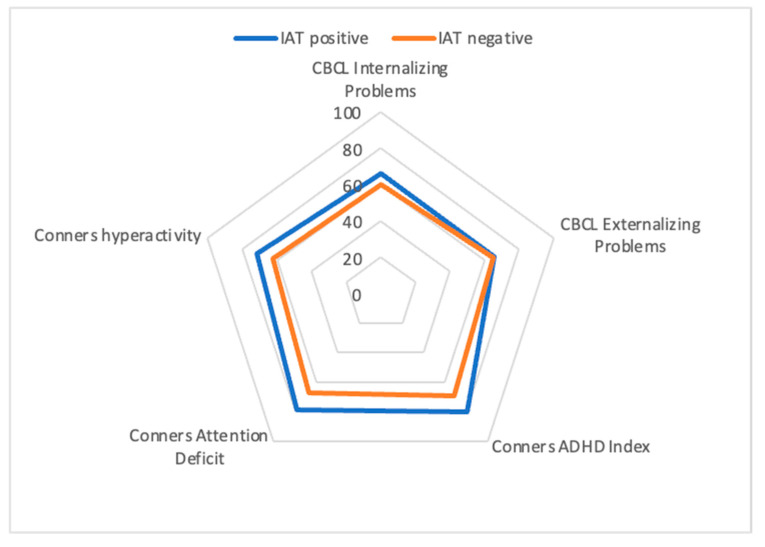
Summary of some of the statistically significant differences in scores between ADHD patients with IAT score ≥ or <50 using the mean scores of Conners’ subscales and CBCL internalizing and externalizing subscales.

**Figure 2 children-09-00428-f002:**
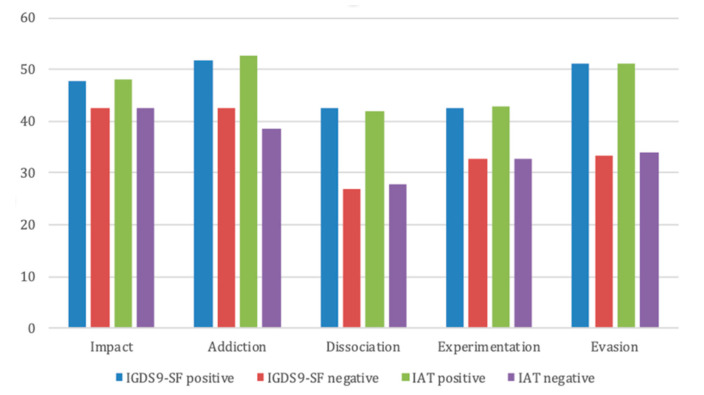
The figure shows the mean scores in the different subscales of the UADI in patients with scores below and above the cut-off at the IGDS9-SF and at the IAT. The clinical cut-off score for the UADI subscales is 50.

**Figure 3 children-09-00428-f003:**
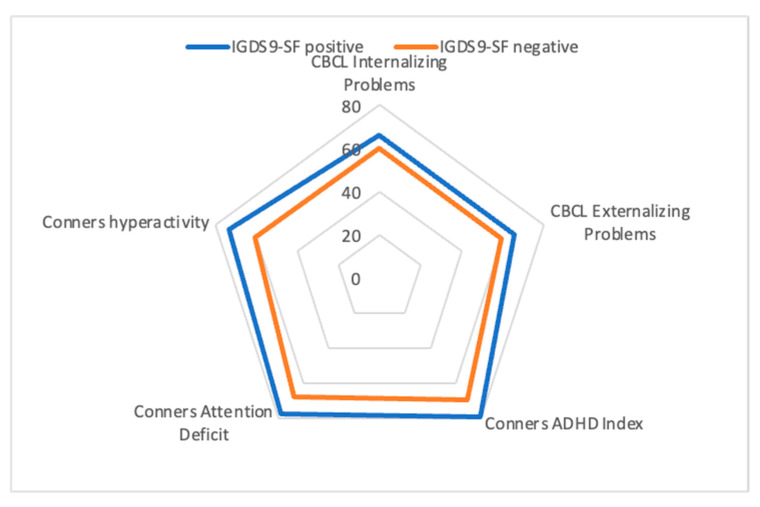
Summary of some of the statistically significant differences in scores between positive and negative IGDS9-SF using the mean scores of Conners’ subscales, CBCL internalizing and externalizing problems.

**Table 1 children-09-00428-t001:** Comparison between subjects scoring above and below the Internet Addiction Test (IAT) cut-off (means and standard deviations). Legend: IAT: Internet Addiction Test; C-GAS: Children’s Global Assessment Scale; CGI-S: Clinical Global Impression-Severity score; CPRS: Conners Parent Rating Scale; CBCL: Child Behavior Checklist; WISC-IV: Wechsler Intelligence Scale for Children—Fourth Edition. *: Statistical significance (*p* < 0.05).

Variables	IAT above the Cut-Off (*N* = 46)		IAT below the Cut-Off (*N* = 62)		*p*-Value
	Mean	SD	Mean	SD	
C-GAS	52.8	6.5	54.9	7.1	0.114
CGI-S	4.1	0.8	3.8	0.7	0.792
CPRS Inattention score	78.7	13.6	67.3	14.1	<0.001 *
CPRS Hyperactivity/Impulsivity score	71.6	17.8	62.3	16.7	0.007 *
CPRS ADHD Index	80.5	12.6	69.1	15.2	<0.001 *
CBCL Internalizing Problems	66.1	9.3	60.2	14.3	0.016 *
CBCL Externalizing Problems	64.8	9.2	60.4	13.1	0.052
CBCL Total score	68.1	8.2	62.2	12.9	0.007 *
CBCL Anxiety/Depression subscale	65.9	9.6	61.6	13.1	0.061
CBCL Withdrawal/Depression subscale	66.9	11.8	61.1	11.9	0.013 *
CBCL Somatic Complaints subscale	60.2	9.1	58.1	11.2	0.292
CBCL Social Problems subscale	67.0	8.9	59.7	12.3	0.001 *
WISC-IV Total IQ	82.8	37.1	76.4	41.7	0.408
WISC-IV Verbal Comprehension Index	101.8	14.7	106.5	14.1	0.094
WISC-IV Visual Perception Index	104.1	15.8	103.9	15.5	0.943
WISC-IV Working Memory Index	87.5	16.3	90.2	14.0	0.350
WISC-IV Processing Speed Index	81.3	14.0	83.9	12.9	0.368

**Table 2 children-09-00428-t002:** Comparison between subjects scoring above and below the Internet Gaming Disorder Scale–Short-Form (IGDS9-SF) cut-off (means and standard deviations). Legend: IGDS9-SF: Internet Gaming Disorder Scale–Short-Form; C-GAS: Children’s Global Assessment Scale; CGI-S: Clinical Global Impression-Severity score; CPRS: Conners Parent Rating Scale; CBCL: Child Behavior Checklist; WISC-IV: Wechsler Intelligence Scale for Children—Fourth Edition. *: Statistical significance (*p* < 0.05).

Variables	IGDS9-SF above Cut-Off (*N* = 48)		IGDS9-SF below Cut-Off (*N* = 60)		*p*-Value
	Mean	SD	Mean	SD	
C-GAS	51.6	6.5	55.9	6.7	0.001 *
CGI-S	4.2	0.8	3.7	0.8	0.001 *
CPRS Inattention Score	77.7	13.5	67.8	14.7	<.0.001 *
CPRS Hyperactivity/Impulsivity Score	73.2	17.8	60.7	15.7	<0.001 *
CPRS ADHD index	79.7	13.0	69.4	15.3	<0.001 *
CBCL Internalizing Problems	66.1	10.3	60.0	13.8	0.012 *
CBCL Externalizing Problems	65.75	9.8	59.5	12.5	0.006 *
CBCL Total score	65.7	9.8	61.8	12.5	0.002 *
CBCL Anxiety/Depression subscale	66.3	10.6	61.1	12.4	0.022 *
CBCL Withdrawal/Depression subscale	67.0	11.9	60.8	11.7	0.009 *
CBCL Somatic Complaints subscale	60.6	9.3	57.6	11.0	0.133
CBCL Social Problems subscale	67.1	10.2	59.4	11.5	<0.001 *
CBCL Thought Problems subscale	65.2	10.9	59.9	12.1	0.022 *
CBCL Attention Problems subscale	71.7	12.1	65.6	13.1	0.015 *
CBCL Rule Breaking Behaviour subscale	69.6	12.2	61.1	12.5	0.001 *
CBCL Aggressive Behaviour subscale	62.3	8.6	58.4	11,004	0.045 *
WISC-IV Total IQ	78.2	39.8	79.9	40.1	0.828
WISC-IV Verbal Comprehension Index	102.0	14.1	106.5	14.5	0.110
WISC-IV Perceptual Reasoning Index	102.2	15.9	105.5	15.2	0.277
WISC-IV Working Memory Index	87.5	16.4	90.2	13.9	0.878
WISC-IV Processing Speed Index	79.7	17.5	85.3	12.2	0.058

## Data Availability

The data presented in this study are available on request from the corresponding author.

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
