# Peer review of "Internet Gaming Disorder in Children and Adolescents with Attention Deficit Hyperactivity Disorder"

_children, 2022, doi:10.3390/children9030428_

Round 1

Reviewer 1 Report

The concept of internet gaming disorder is novel.  The study has indeed provided an insight on association of IGD with ADHD.  The authors indicated that the same size of clinical population is mostly male (96%) whereas the healthy population is both male and female (78% and 22% respectively). 

  1. It is suggestable for the authors to explain the basis of sample size calculation.
  2. Do the difference in sample size of male and female (equal distribution) between clinical and healthy group inflence of the results of the study.  If so, how did the authors approach this problem?

Author Response

The basis of sample size calculation has been provided in the Method Section and in the Statistical analyses section.

Regarding the possible influence of the differences in gender ratio between normal control group and the clinical sample, the paucity of females did not allow us to explore possible gender specificities. This issue is certainly important, it deserves specific research and the lack of information in our study has been included in the limitations.

Reviewer 2 Report

1.- Introduction. The bibliography must be updated. Both addiction to new technologies and ADHD are very topical issues on which there is important recent research, which is not cited.

2.- Methodology. At the time the researchers collect the sample, dozens of studies highlight that the use of new technologies has increased markedly in the child and adolescent population. In addition, the situation we are experiencing at the international level has led to an increase in the level of state anxiety. There are also studies that affirm that, in the population with neurodevelopmental disorders such as ADHD, those previously described, it is even more pressing. Have they taken it into account to avoid the strange variable? excuse me? I think they should highlight it in the introduction and include, as an improvement, a new sample once this variable, which directly affects the study population, has been controlled.

3.- Measures: The strategy followed to collect the sample information seemed very solid to me. I congratulate the research team, since both the most significant variables in the research, addiction and ADHD, are masterfully collected. I recognize as especially costly, but at the same time a very good decision, the use of WISC-IV for the selection of the clinical group diagnosed with ADHD.

4.- In Methodology important changes must be made, we include a pdf with specific marks and guidelines. In this pdf we also highlight other errors that we have found.

5.- Discussion and conclusions of the study very well developed. Honestly, I think it's a great work, very transferable to clinical practice and that provides useful information to science. Despite this, I reaffirm that they must improve the previous systematic review, and take into account "the state of the art" in relation to the subject under investigation.

6.- English would require revision by a native speaker.

Author Response

In the Introduction new references have been included

Matherial and Methods: All the suggestions included in the (very helpful) pdf have been included in the revised version, including the psychychometric properties of the measures and the inclusion of the effect sizes. A new section on Procedure has been added, according to the reviewer’s suggestion

The first sentence has been rephrased, as well as the second paragraph.

Statistical analyses: further information has been added in this section

Tables: the mistakes have been corrected

The Discussion has been developed, including new theoretical considerations

English has been revised by a native speaker

Round 2

Reviewer 2 Report

I consider that the modifications carried out are sufficient for the work to be published in the journal. My congratulations to the authors.